# Blind demixing methods for recovering dense neuronal morphology from barcode imaging data

Shuonan Chen[1,2,3,4,5,6]*, Jackson Loper[1,2,3,4,5,7], Pengcheng Zhou[8,9], Liam Paninski[1,2,3,4,5,7]

**1** Mortimer B. Zuckerman Mind Brain Behavior Institute, Columbia University, New York, New York, United States of America, **2** Department of Statistics, Columbia University, New York, New York, United States of America, **3** Center for Theoretical Neuroscience, Columbia University, New York, New York, United States of America, **4** Grossman Center for the Statistics of Mind, Columbia University, New York, New York, United States of America, **5** Department of Neuroscience, Columbia University, New York, New York, United States of America, **6** Department of Systems Biology, Columbia University, New York, New York, United States of America, **7** Data Science Institute, Columbia University, New York, New York, United States of America, **8** Faculty of Life and Health Sciences, Shenzhen Institute of Advanced Technology, Chinese Academy of Sciences, Shenzhen, China, **9** The Brain Cognition and Brain Disease Institute, Shenzhen Institute of Advanced Technology, Chinese Academy of Sciences, Shenzhen, China

* sc4417@cumc.columbia.edu

**Data Availability Statement:** The data underlying the results presented in the study is available from https://github.com/jacksonloper/bardensr.

**Funding:** This work was supported by the National Institutes of Health (1U19NS107613 to L.P.),

## Abstract

Cellular barcoding methods offer the exciting possibility of 'infinite-pseudocolor' anatomical reconstruction—i.e., assigning each neuron its own random unique barcoded 'pseudocolor,' and then using these pseudocolors to trace the microanatomy of each neuron. Here we use simulations, based on densely-reconstructed electron microscopy microanatomy, with signal structure matched to real barcoding data, to quantify the feasibility of this procedure. We develop a new blind demixing approach to recover the barcodes that label each neuron, and validate this method on real data with known barcodes. We also develop a neural network which uses the recovered barcodes to reconstruct the neuronal morphology from the observed fluorescence imaging data, 'connecting the dots' between discontiguous barcode amplicon signals. We find that accurate recovery should be feasible, provided that the barcode signal density is sufficiently high. This study suggests the possibility of mapping the morphology and projection pattern of many individual neurons simultaneously, at high resolution and at large scale, via conventional light microscopy.

## Author summary

*In situ* barcode sequencing allows us to simultaneously locate many neurons in intact brain tissues, albeit at modest spatial resolution. By increasing the barcode density, high-resolution neuronal morphology reconstruction from such data might be possible. Here we use simulations to study this possibility, while addressing the computational challenges in analyzing such data. We developed a novel blind demixing method that uses fluorescent images and identifies the unknown barcodes used to label the neurons with high

IARPA MICrONS (D16PC0003 to L.P.), and the
Chan Zuckerberg Initiative (2018-183188 to L.P.).
The funders had no role in study design, data
collection and analysis, decision to publish, or
preparation of the manuscript.

accuracy. Further, we developed a neural network which can reconstruct the morphology
for these labeled neurons from the observed 'pointilistic' imaging data. We show that
under both high- and low-resolution optical settings, our methods can successfully extract
the morphologies for many labeled neurons. The results from this theoretical study sug-
gest that it may be feasible to map the morphology and projection pattern of many indi-
vidual neurons simultaneously, at high resolution and at large scale, via conventional light
microscopy.

## 1 Introduction

Neuroscientists have long dreamed of obtaining simultaneous maps of the morphology of
every neuron in a mammalian brain [1, 2]. The ability to perform this very high-throughput
neuron-tracing would enable better understanding of brain development, neural circuit struc-
ture, and the diversity of morphologically-defined cell types [3, 4]. In order to map the mor-
phology at the scale of a mammalian brain, an ideal experiment would trace a large number of
neurons simultaneously over a large brain region with high imaging resolution.

Current experimental methods for neuronal tracing can be placed along a spectrum span-
ning two disparate spatial scales. On the one hand, techniques based on Electron Microscopy
(EM) obtain nanometer-level resolution within small regions. On the other hand, light micros-
copy and molecular techniques can map morphology over whole-brain scales, albeit with rela-
tively low spatial resolution and/or limitations on the number of neurons that can be mapped
simultaneously. Fig 1 sketches the current landscape, and highlights a gap in the state of the
art: mapping many neurons simultaneously, with high resolution, over large fields of view (or
potentially even the whole brain).

Molecular barcoding methods may be able to fill the gap [5, 6] with 'infinite-color Brain-
bow' experiments: each cell is labeled with a unique random barcode which can be visualized
through conventional light microscopy. If we think of each barcode as providing a unique
'pseudocolor' for each cell, then conceptually this method 'colors' the brain with a potentially
unbounded number of random colors, generalizing the original Brainbow approach [7], which
can trace long-range neuronal morphology but was limited to a relatively small number of ran-
dom colors (though see e.g. [8] for more recent developments).

Here we use simulations to investigate a class of methods we dub Spatial Transcriptomics-
based Infinite-color Brainbow Experiments (STIBEs). Can such experiments obtain precise
morphological reconstruction? Can we potentially obtain long-range tracing for many neu-
rons simultaneously, if the experiment is done at a sufficienctly large scale? This class of experi-
ments uses techniques from the spatial transcriptomics community to label cells with unique
fluorescent barcodes, in-situ; the BAR-seq method is a representative example [5]. STIBEs
work by creating a collection of transcripts containing barcode sequences; these transcripts are
amplified into target molecules known as 'amplicons,' and the barcode associated with each
amplicon can be identified over a series of rounds of imaging. In an ideal STIBE, all amplicons
within a given cell carry the same barcode, no two cells carry amplicons with the same barcode,
and amplicons are only present within cells. By identifying collections of amplicons which
carry the same barcode, one can thus trace many neurons across the entire brain.

Previous experiments have already demonstrated that STIBEs can be used to recover the
locations of somas and projection sites of many cells (at lower spatial resolution) [5, 9]. In con-
trast, here we focus on STIBEs that could achieve micron-resolution morphological recon-
struction of many cells simultaneously, potentially across the whole brain. This would be a

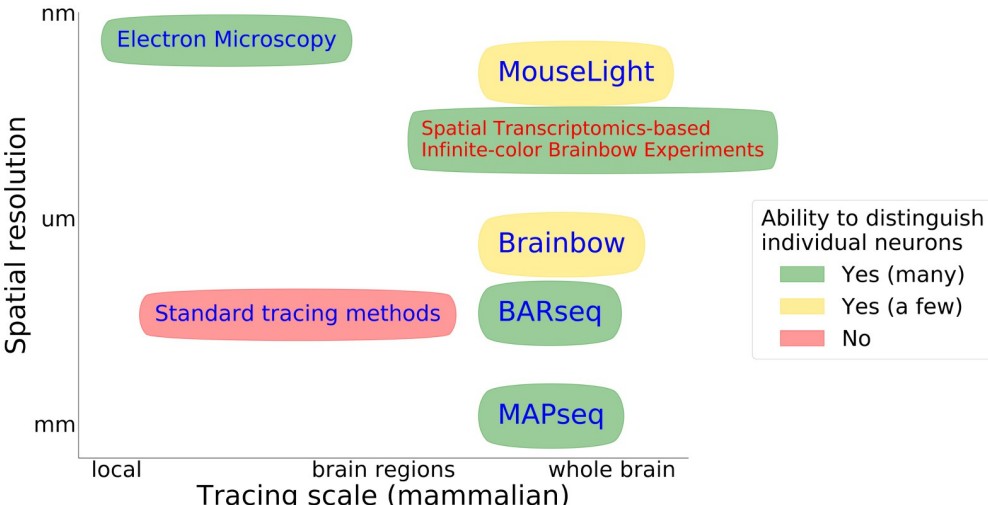

**Fig 1. Some existing methods for neuronal morphology mapping.** Here we non-exhaustively summarize the landscape of existing neuronal tracing methods. The horizontal axis indicates the typical scale of these neuronal tracing methods, ranging from local tracing to mapping axonal projections across the whole brain; the vertical axis indicates optical resolution in the typical images obtained from these methods. The ellipse color represents the number of neurons that can be traced and distinguished; red indicates that the method cannot readily distinguish between spatially nearby neurons, yellow indicates that the method can be used to trace several individual neurons, and green indicates the possibility of simultaneously mapping thousands or millions of neurons. Here 'standard tracing methods' refer to approaches such as Golgi staining and viral tracers; cf. Section 2 for more details. Note that many of these methods rely on conventional optical microscopes, and their resolution can potentially be increased via expansion microscopy and/or super resolution microscopy, though this may incur additional experimental costs. In this paper we focus on the possibility of extending molecular barcoding methods by increasing the signal density to allow for more fine-grained morphological reconstructions; we call such experiments Spatial Transcriptomics-based Infinite-color Brainbow Experiments (STIBEs).

valuable advance for several reasons. First, cell morphology measured at this scale provides a great deal of information about cell type in different layers and regions of the brain [10, 11]. Second, although micron-level resolution does not permit exact connectome reconstruction, it does yield a matrix of *potential connections* between neurons (i.e., which cells might be connected to each other). Such matrices would be invaluable for downstream applications such as functional connectivity mapping (by providing constraints on possible connections) [12] or cell type inference [13–15].

To accurately simulate the imagestacks that would be generated by STIBEs, we use two sources of existing data. First, we use densely-reconstructed EM data to give us the shapes and spatial relationships between axons and dendrites in a small cortical volume. Second, we replicate the noise and signal structure we might expect, based on results from current cellular barcoding experiments [5, 6]. We investigate how varying parameters of STIBEs would affect the accuracy of neuronal tracing; for example, we consider various imaging resolutions and various densities of the amplicons present within cells. We find that selecting an appropriate density of amplicons is a balancing act: we cannot recover precise morphological reconstructions without sufficient density, but very high density can lead to incorrect amplicon identification along thin processes due to limited optical resolution.

Our main technical contribution here is to develop improved demixing algorithms that enable improved barcode recovery in the high-density regime, leading in turn to improved morphological recovery. These algorithms are based on the BarDensr model of spatial transcriptomics data [16] (see also [17], as well as [18, 19]), which was originally developed to

detect barcode signals from spatial transcriptomics images when the barcode library is known. We have extended this method to a more challenging 'blind demixing' setting, where we iteratively learn the barcodes directly from the available image data, so that it is applicable to STIBEs where the barcode library is unknown or intractably large. We further developed a Convolutional Neural Network (CNN) model to reconstruct morphology for each demixed barcode, 'connecting the dots' between the discontiguous fluorescent signals produced by amplicons.

In what follows, we contextualize STIBEs among the family of other neuronal tracing methods, describe our methods for simulation and image analysis, and finally report our findings on the feasibility of STIBEs for the morphological reconstruction of neurons.

## 2 Related work

The history of neuronal tracing techniques is long and complex [20]; here we give a brief overview of some techniques most directly relevant to our approach. Fig 1 presents a visual overview.

At one extreme of the precision-scale spectrum, Electron Microscopy (EM) is the state-of-the-art tool for visualizing a small region of brain with very high precision [14, 15, 21–25]. However, it is not yet suitable for dense mapping of long-range neuronal projections: the imaging (and computational segmentation) process for these experiments is challenging, and it is not yet feasible to use EM to densely trace many neurons across the mammalian brain. It may be possible to refine EM techniques to overcome these challenges, but this is not our focus in this paper (cf. [2] for further discussion).

At the opposite extreme of the precision-scale spectrum, we have standard tracing approaches utilizing light microscopy. Despite significant effort [26], it remains infeasible to segment and trace thousands of densely-packed individual neurons visualized with a single tracing marker. The 'MouseLight' project [27] is a modern exemplar of this 'classical' approach. This is a recently developed platform for tracing the axonal arbor structure of individual neurons, built upon two-photon microscopy and viral labeling of a sparse set of neurons, allowing long-range neuronal tracing. Unfortunately, the number of neurons that can be traced at once is currently limited to a hundred per brain sample [28] (though more advanced microscopy technology, such as super-resolution microscopy [29–33], and/or Expansion Microscopy [34–37], could potentially be deployed to simultaneously trace more individual neurons with higher resolution).

'Brainbow' methods [7, 38] introduce random combinations of fluorescent markers to facilitate tracing of multiple cells. The number of neurons that can be uniquely labeled and identified by the original method was limited to several hundreds, because of limitations on the number of distinct colors that can be generated and distinguished. Recent studies have focused on reducing these bottlenecks [8, 39, 40], and developing improved computational methods [41, 42].

More recently, molecular barcoding methods have been developed to effectively remove this bottleneck on the number of distinct unique 'color' labels that can be assigned and read from different cells. Multiplexed Analysis of Projections by Sequencing (MAPseq) is one early example of this approach, developed to study long-range axonal projection patterns [9]; however, the spatial precision of this approach was not sufficient to capture neuronal morphology. More recently, "barcoded anatomy resolved by sequencing" (BARseq) [5, 6] combines MAPseq and *in situ* sequencing technology to obtain higher spatial resolution. Instead of micro-dissecting the brain tissue, this method uses fluorescent microscopy to detect the barcode signal from the intact tissue, similar to recent spatial transcriptomics methods [43, 44]. In theory,

BARseq should be capable of satisfying all the desirable features of neuronal tracing: uniquely labelling large numbers of individual neurons, representing them with high spatial precision, and tracing them across long distances through the brain. Spatial transcriptomics technology is advancing quickly; for example, [45] recently achieved single-cell-resolution transcriptomic assays of an entire embryo. As this technology advances, techniques like BARseq can only become more effective. However, to date, this technology has only been used with a relatively small number of amplicons per cell, prohibiting accurate morphological reconstruction (c.f. Brainbow images, where signal within cells tends to be much more uniform and less 'pointilistic,' leading to a different class of segmentation problems). In theory, there is nothing preventing new experiments with higher density, to achieve higher-resolution morphology mapping; this paper will explore this idea systematically.

Finally, we note the related paper [46]; this previous simulation work focused on a different experimental context, specifically expansion microscopy and cell membrane staining with fluorescent markers. In the current work we are interested in exploring the frontier of morphology mapping using only molecular barcoding images and conventional light microscopy. Introducing expansion microscopy and additional fluorescent labels could potentially improve the resolution of the methods studied here, at the expense of additional experimental complexity.

## 3 Methods summary

We here summarize the three main procedures used in this work; full details can be found in S1A Appendix.

### 3.1 Data simulation

In order to investigate feasible experimental conditions for densely reconstructing neuronal morphology, we simulated a variety of STIBEs. The simulation process is illustrated in Fig 2. We started with densely segmented EM data [14, 15, 47]. Next we generated a random barcode library, assigning random barcodes to each cell in the field of view (FOV). We simulated amplicon locations according to a homogeneous Poisson process within the voxelized support of each cell. The final imagestack was generated by assigning a colored spot to each barcode location in each imaging frame, and then pushing the resulting high-resolution 3D 'clean' data through an imaging model that includes blurring with a point spread function, sampling with a lower-resolution voxelization, and adding imaging noise, to obtain the simulated observed imagestack.

### 3.2 Barcode estimation

To use a STIBE for morphological reconstruction, one must estimate the barcodes present in a FOV (each barcode corresponding to one cell) and the locations of each amplicon corresponding to each barcode (these amplicon locations pointilistically trace the morphology of each cell). The barcode library used to infect the neurons is unknown in such data. In theory, one could sequence the viral library, but this may yield a set of barcodes which is too large to be useful [48]. For the purpose of morphology reconstruction—where only a small number of cellular barcodes exist in a region of interest—we must develop an approach to 'learn' the local barcode library from the images themselves. We found that there are effectively two distinct regimes for barcode and amplicon recovery. In the 'sparse' regime, where imaging resolution is high and the amplicon density is low, the signal in most imaging voxels is dominated by at most one amplicon. Thus we can estimate the barcode library simply by searching for bright, 'clean' imaging voxels displaying a single amplicon signal to estimate the amplicon locations

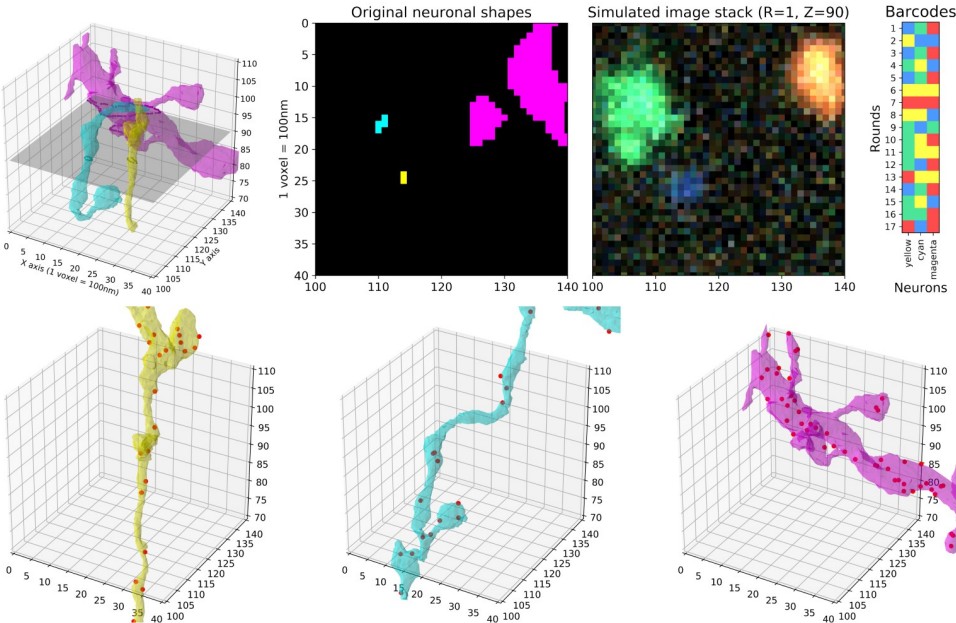

**Fig 2. Simulation process illustration. Top left**: Three neuronal segments in a small region ($40 \times 40 \times 40$ voxels with $100 \times 100 \times 100$ *nm* voxel size). The original EM data is much more densely packed than what is shown here; for illustration purposes, we only show three neurons among many. **Top middle**: A 2D slice with each neuronal segment uniquely colored (left) and the simulated imagestack at the same plane for the first sequencing round (right). The imagestack colors do not correspond to neural segment colors, but rather to the corresponding fluorescent barcodes (top right). A video corresponding to this plot, showing multiple z-planes, can be found at this link. **Bottom**: Amplicons are uniformly simulated within neuronal segment volumes; simulated amplicons are shown as red dots.

using the resulting barcode library. In the 'dense' regime (with high amplicon density and/or low imaging resolution) it is harder to find voxels dominated by a single amplicon, and the simple approach described above breaks down. Instead, we have found that an iterative constrained non-negative matrix factorization approach is more effective: given an initial (incomplete) estimate of the barcode library, we estimate the corresponding amplicon locations, then subtract away the estimated signal corresponding to these amplicons. This sparsens the remaining image, making it easier to detect more barcodes that might have been obscured in previous iterations. After augmenting the barcode library with these previously undetected barcodes, we can re-estimate the amplicon locations and iterate between these two steps (updating the barcode library and estimating amplicon locations in alternating fashion) until a convergence criterion is satisfied. This approach takes full advantage of our knowledge of the special structure of this blind-demixing problem; classic blind-demixing approaches (e.g. regular non-negative matrix factorization) that do not exploit this problem structure do not perform well here.

### 3.3 Amplicon estimation and morphological reconstruction

Given an estimated barcode library, our primary goal is to reconstruct the morphology of each neuron labeled by a barcode in this library. Our starting point in this endeavor is the 'evidence tensor' (see S1 Appendix for a precise definition), which summarizes our confidence about the presence of each barcode at each voxel location. This evidence tensor is used as the input to two algorithms: alphashape [49] and Convolutional Neural Networks (CNNs), which are trained to estimate the shape of each neuron from the evidence tensor.

## 4 Results

Below we report our ability to recover barcodes and reconstruct morphology in two different simulation regimes: the 'high-resolution, dense-expression' regime, in which every neuron is labeled and imaged with sub-micron resolution; and the 'low-resolution, sparse-expression' regime, in which only about 1% of neurons are labeled and imaged with micron-resolution imaging. The first regime ('high-resolution simulation' for short) represents an ideal setting. For this regime, we also used real experimental data to test if the barcode recovery method works on real-world data with a known set of barcodes. Experiments in the second regime ('low-resolution simulation' for short) would be considerably cheaper to perform, with lower optical resolution facilitating faster image acquisition over large FOVs; however, the resolution in this simulation is low enough that many cellular barcodes may occupy a single voxel. To enable accurate recovery in this regime we must also assume that the cell capture efficiency is relatively low: i.e., only a fraction of neurons express a barcode.

### 4.1 High-resolution, dense-expression simulations

In our first set of simulations, we assume diffraction-limited imaging with an isotropic point spread function (PSF). We further assume that the viral infection has a perfect efficiency; in this extreme case, all the neurons captured by EM in this brain region are labeled with a unique barcode. The simulation process is illustrated in Fig 2. With these assumptions fixed, we investigated a range of experimental parameters, varying the amplicon density as a key parameter of interest. We denote this density by λ and measure it with units of amplicons per cubic micron of neuronal volume. We investigated densities from 1 amplicon per cubic micron (i.e., λ = 1) to 200 amplicons per cubic micron (i.e., λ = 200). For each density variant we investigated our ability to recover barcodes. The simulation results using three of these densities (1, 10 and 100 amplicons/$\mu m^3$) are summarized in Fig 3. More details on the simulation process can be found in S1A.1 Appendix.

**4.1.1 Barcode estimation.**   To estimate the barcode library present in a STIBE, we take advantage of a special structure in the barcodes of typical spatial transcriptomics experiments: each fluorescent barcode is designed so that exactly one channel is active in each imaging round. This suggests a simple approach for recovering the barcode library: search for voxels where one channel is much brighter than all others in every round. This approach is successful in some cases. Barcode discovery using this simple approach is easiest with a medium density of amplicons. In the low-density case, with λ = 1 amplicon/$\mu m^3$, we were able to find 847 out

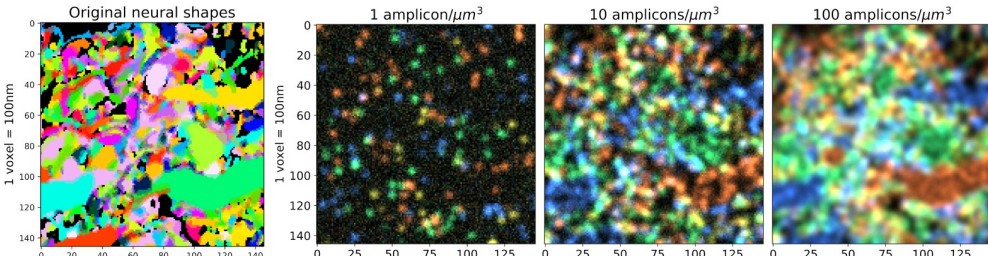

**Fig 3. High-resolution, dense-expression simulation with varying amplicon densities. Left**: 2D slice of the original neuronal shapes; each neuron is visualized with a different random color. Partial overlapping due to point-spread blur can be seen from the mixing of the colors on the boundary of the neurons. **Right**: Corresponding slice of the simulated imagestack, considering several different density values. Note that the imagestack colors do not correspond to the neuronal visualization colors on the left; the imagestack colors arise from their corresponding barcodes. A video showing multiple z-planes can be found at this link.

of 975 barcodes present in tissue using this naïve approach, representing a detection rate of 86.87%, with 0 false positives (however, as we discuss later, our ability to reconstruct neuronal morphology in this low-density case is quite limited). Increasing the density to 10 transcripts per cubic micron ($\lambda = 10$ amplicons/$\mu m^3$), we are able to detect 945 barcodes (96.92% detection rate) with 2 false positives. Note that barcode recovery rates drop in the very low amplicon density regime, since some small cells in the FOV may not express enough amplicons for reliable detection. (Of course in practice a user may wish to adjust the detection threshold according to their own needs, since the relative cost of a false negative versus false positive will likely vary across different experimental conditions).

As shown in Fig 4, even with medium density the signal can be significantly mixed; that is, a single voxel can display signal from multiple cells, even under the assumption of high-resolution imaging system. When this occurs, correctly identifying the barcodes becomes

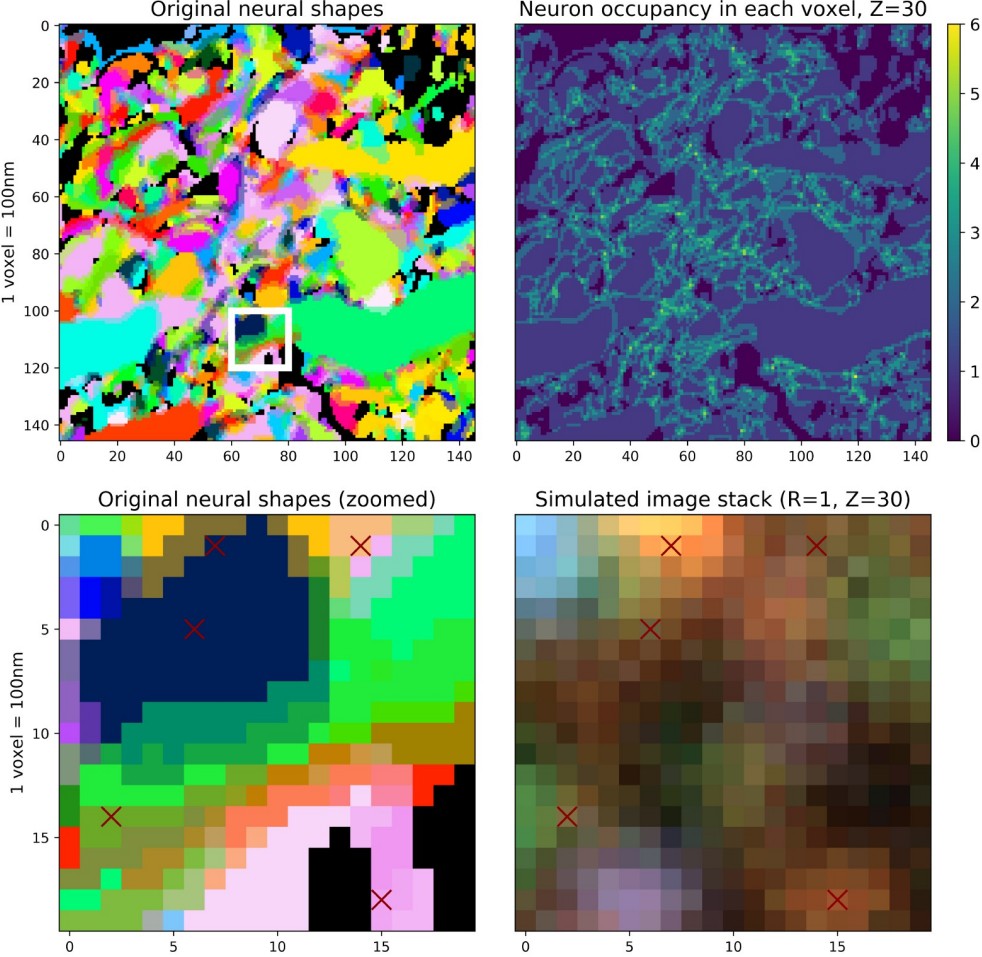

**Fig 4. Limited optical resolution leads to signal mixing between neighboring neurons even in the high-resolution setting. Top left**: Neuronal shapes in high-resolution, dense-expression data, as seen in Fig 3. **Top right**: The number of distinct neurons that contribute signal to each voxel. In some densely-packed regions, each voxel might mix signal from as many as six cells, due to optical blur and the voxelization process. **Bottom left**: Zoomed-in 20 × 20 region of the original neuronal shapes (the white rectangular region from the top left panel). **Bottom right**: Simulated imagestack in the same region, illustrating the challenge of uniquely assigning amplicons to neurons ($\lambda = 10$ amplicons/$\mu m^3$ here). Simulated amplicon locations within this z-plane are shown with red crosses; other amplicons centered outside of this z-plane contribute additional signal.

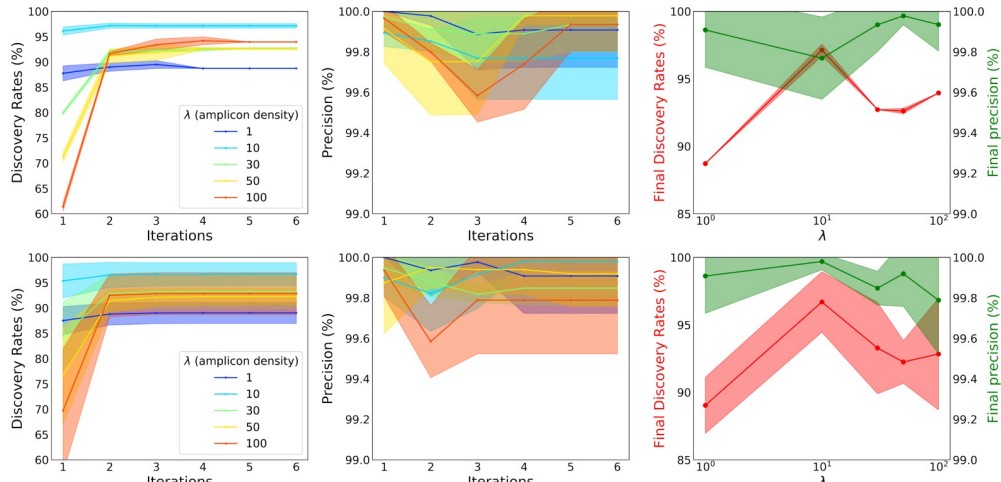

**Fig 5. An iterative non-negative matrix factorization approach enables barcode discovery in the high-amplicon-density regime for high-resolution, dense-expression simulations. Top**: five datasets are randomly simulated from a single EM volume and performances are summarized here. The left panel shows barcode discovery rate, as a fraction of the 975 neurons present in the original EM volume. The middle panel shows the precision at each iteration, which is the proportion of True Positives among all the positive detections. These two panels show these metrics evaluated on STIBEs with various densities of amplicons (λ, colored lines) analyzed using different numbers of iterations. The shaded regions for each line indicate the standard deviation. We see that multiple iterations improve recovery accuracy when the density of amplicons is high. The right panel shows the final discovery rate (vertical axis on the left, red) as well as the final precision (vertical axis on the right, green) after six iterations, as a function of amplicon density (at log-scale, horizontal axis). Recovery accuracy peaks at λ values around 10 amplicons/$\mu m^3$, where amplicons are neither too dense (where too much optical overlap leads to decreases in recovery accuracy) nor too sparse (where there is simply not enough information to recover some barcode identities). **Bottom**: five datasets are randomly simulated from five different EM volumes and the performances are summarized here. The plots are arranged in the same way as above. Here we see a very similar trends, albeit with larger variations across the replicates, confirming the results generalize across multiple FOVs. These plots further confirmed that 10 amplicons/$\mu m^3$ is an optimal value for this iterative barcode discovery task, under these simulated imaging conditions.

challenging even manually. Increasing the density further to 100 transcripts per cubic micron (λ = 100 amplicons/$\mu m^3$), this problem becomes severe: the naïve method can detect only 604 barcodes, representing a detection rate of 61.95%, with zero false positives.

In short, we found the naïve barcode recovery method inadequate when the amplicon density is high, even under idealistic assumptions on optical resolution. In order to recover most of the barcodes, we need to demix the accumulated signal emitted by many barcodes contributing to a single voxel. We adopted a novel iterative approach based on non-negative matrix factorization, detailed in S1A.2.2 Appendix and Algorithm 3. The results of this iterative barcode discovery approach under different amplicon densities are shown in Fig 5. Here, we generated two types of datasets, shown on the top and the bottom panels respectively. The first type of dataset was simulated from a single EM volume, and the second type was simulated from different EM volumes. For the first type, we simulated five random datasets from a single EM volume, using a wide range of amplicon densities. In this case, the randomness applies to both codebook and spot locations within the cell region, but all the datasets share the same ground-truth neuronal regions. For the second type, five different EM subvolumes are used to generate five random datasets. In this case, each of these five datasets has a different ground-truth neuronal region. In all cases we chose the parameters to ensure at most five total false positives throughout the iterations, to maintain a high level of precision throughout. This was done by increasing the signal_control value (see S1A.2.1 Appendix) until the false positives stayed below five throughout all iterations; a high value of this parameter leads to a more

conservative discoveries. We found that 2–3 iterations consistently improve performance, yielding 30% more detections when the amplicon density is very high ($\lambda$ = 100 amplicons/ $\mu m^3$).

In summary, in our high-resolution setting, we can accurately detect the barcode library using the naïve approach when the signal density is relatively low. On the other hand, when the signal density is high, we can still recover most of the barcodes correctly by using the iterative approach.

**4.1.2 Real experimental data barcode estimation.** In order to test our method on real-world data, we used benchmarking data from a previous study [16], where the barcodes are used to identify genes rather than cells. In this experiment, the barcodes are known beforehand; we can use these barcodes as the ground-truth codebook, and attempt to recover this codebook. Note that because the barcodes correspond to genes instead of cells, we are not able to use this dataset to benchmark cellular morphology reconstruction results. Fig 6 (top panel) displays a comparison of this real experimental data against our simulated data; we see a reasonable correspondence between the simulated and real data here, in terms of both the expression density and image signal-to-noise ratio.

In this ground-truth codebook, we have 79 barcodes used to target mRNAs, and 70 of them had at least one amplicon in the images (the number of amplicons are estimated using BarDensr [16], which utilizes the known barcode library). As in the simulations above, we applied the naïve barcode discovery approach first. Using the naïve approach we could only

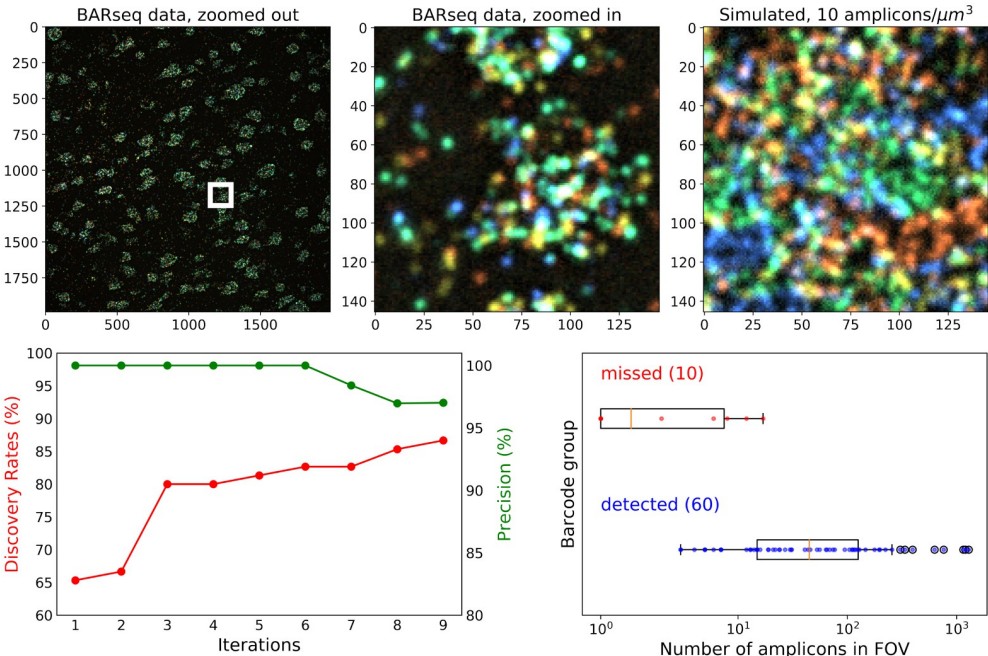

**Fig 6. Barcode iterative discovery for real experimental data [16]. Top**: An example image from the first round of imaging is shown on the left panel. The middle panel is a zoomed-in view of the white rectangle on the left. On the right panel, we showed the simulated data with 10 amplicons/$\mu$m³ density to show that our simulation corresponds reasonably well to the real experimental data. **Bottom**: The result of iterative barcode discovery on the entire FOV shown on the left top panel. The discovery rate (red) and precision (green) over each iteration are shown on the left plot. We also investigated if there is an association between the barcode detection accuracy and the number of amplicons in the image that belong to that barcode. On the right panel, we show that all missed barcodes are associated with small numbers of amplicons; barcode detection can be achieved as long as we have a sufficiently large number of amplicons present in the FOV.

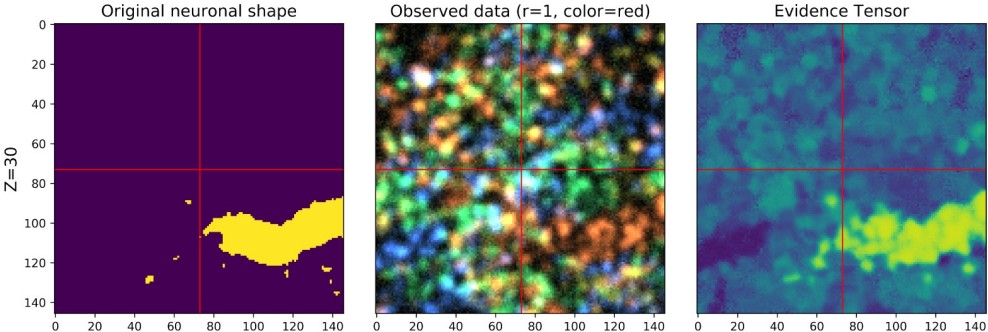

**Fig 7. The evidence tensor is used to match barcodes to voxels.** The ground-truth morphology at a selected z-plane for an example neuron is shown on the left; an imagestack from one round at the corresponding z-plane ($\lambda = 10$ amplicons/$\mu m^3$) is shown in the middle; on the right, the corresponding z-plane of the evidence tensor for the barcode corresponding to the neuron on the left is shown. Note that the evidence tensor is bright around the locations of the underlying neuron, as desired. A video of this image (across z-planes) can be found at this link.

detect 65.3% of all the barcodes with at least one amplicon. Using the iterative approach, the detection rate increased to 86.7%, as shown in Fig 6 bottom left. (For details on the barcodes estimated using real data, see S1 A.2.3 Appendix) The few missed barcodes here (i.e., False Negatives) tend to correspond to targets with relatively small numbers of amplicons in the FOV (Fig 6 bottom right). Two examples of missed barcodes are shown in S2 Fig. Note that this experiment was performed with only six imaging rounds (rather than 17 imaging rounds used in our simulation settings) and these shorter barcode sequences lead to higher error rates than seen in the simulated results.

**4.1.3 Amplicon estimation and morphological reconstruction.** In order to estimate the morphology corresponding to each recovered barcode, we start by estimating the density of each barcode at each voxel; we refer to this estimate as the 'evidence tensor' (cf. S1A.3 Appendix). Note that this 'evidence tensor' is computed using barcodes recovered from the imagestack (rather than the ground-truth barcodes used for creating the simulations). Fig 7 compares the original neuronal shape and the corresponding evidence tensor; we see that the evidence tensor can roughly capture the shape of the original neuron.

We also investigated neuronal tracing using the estimated evidence tensor with low, medium and high-density variations in Fig 8. The binarized evidence tensors computed from medium-density variation can capture the original morphology relatively well, by correctly identifying the amplicon locations (the third row). However, those learned from low-density variations cannot capture the original morphology as clearly (the second row). The rough path of the neuron can still be identified, but spines are generally missed from dendrites and submicron resolution is not achieved. The high-density variation results were similar to the medium-density variation, but with more continuous shape reconstruction (the bottom row). However, they tend to miss the detailed shapes, as well as a portion of the barcodes, due to overlaps of the simulated amplicons in such high-density images. Note here that the evidence tensors are not created to recover the individual amplicon locations but rather to aid in high fidelity reconstructions of the neuronal morphology. High values of this tensor indicates *potential* amplicon locations, but the set of locations where such high values are achieved is generally larger than the set of locations where amplicons reside (due to blurring induced in the imaging process).

Next we want to estimate the morphology of each neuron from the evidence tensor. We experimented with two approaches here. The first approach applies the alphashape algorithm

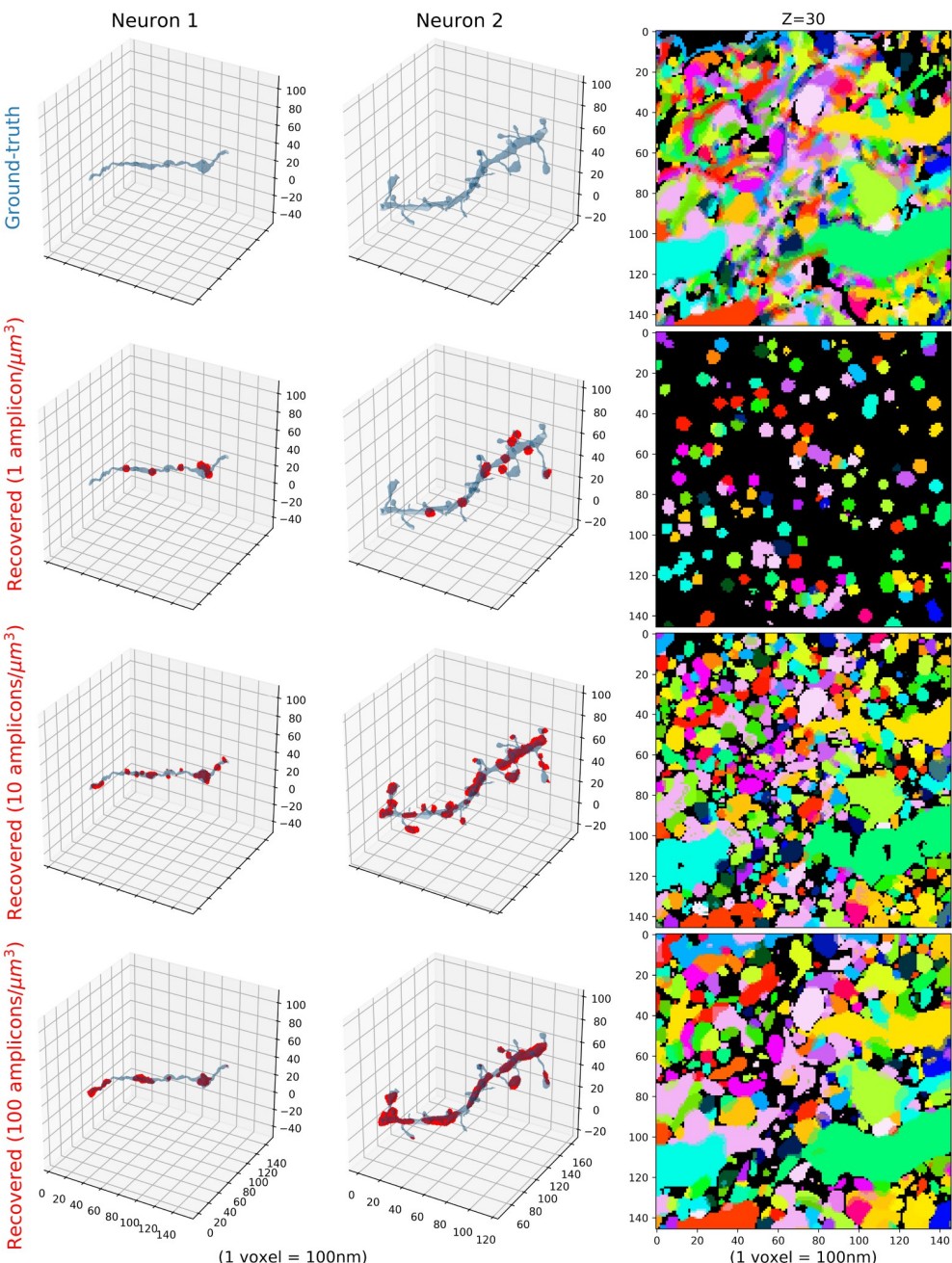

**Fig 8. Estimated amplicon locations (binarized evidence tensor) from high-resolution, dense-expression simulations, with 1, 10 and 100 amplicons per cubic micron.** Given an imagestack from a STIBE, how well can we estimate the amplicon regions and can we roughly recover the morphology of individual neurons based on these regions? The first column visualizes one cell, the second visualizes another, both shown in blue shaded regions. The first row indicates ground-truth morphology of the cells. The second row shows our ability to locate the amplicon regions inside the cells using a high-resolution STIBE with λ = 1 amplicon/$\mu m^3$; identified locations, which are estimated using the binarized evidence tensor with threshold 0.7 (see S1 Fig for details on the choice of this threshold), are shown in red in the first two columns, and via unique colors in the third column. Note that these regions are often larger than the actual size of the amplicons because of point-spread effects during imaging. The final two rows visualize the same information for a STIBE with λ = 10 amplicons/$\mu m^3$ and λ = 100 amplicons/$\mu m^3$. **Right column**: On top, we visualize many cells in a single slice of tissue, each indicated with a unique color. The following rows show the recovered neuronal shapes based on the binarized evidence vector. Colors match the right top panel. A video showing multiple z-slices corresponding to the right panel can be found at this link.

[49] to a thresholded version of the evidence tensor; this enables better reconstruction of thin portions of the neuron by incorporating our knowledge that neurons occupy contiguous regions in space. Alphashape is a classical method to reconstruct morphology from a point cloud, obtained by computing Delaunay simplices and removing simplices with large circum-radii. Fig 9 demonstrates that alphashape is able to estimate the continuous shape of neurons by 'connecting the dots' in the evidence tensor. However, it tends to over-estimate the shape, perhaps because alphashapes can only construct linear connections between pairs of detected

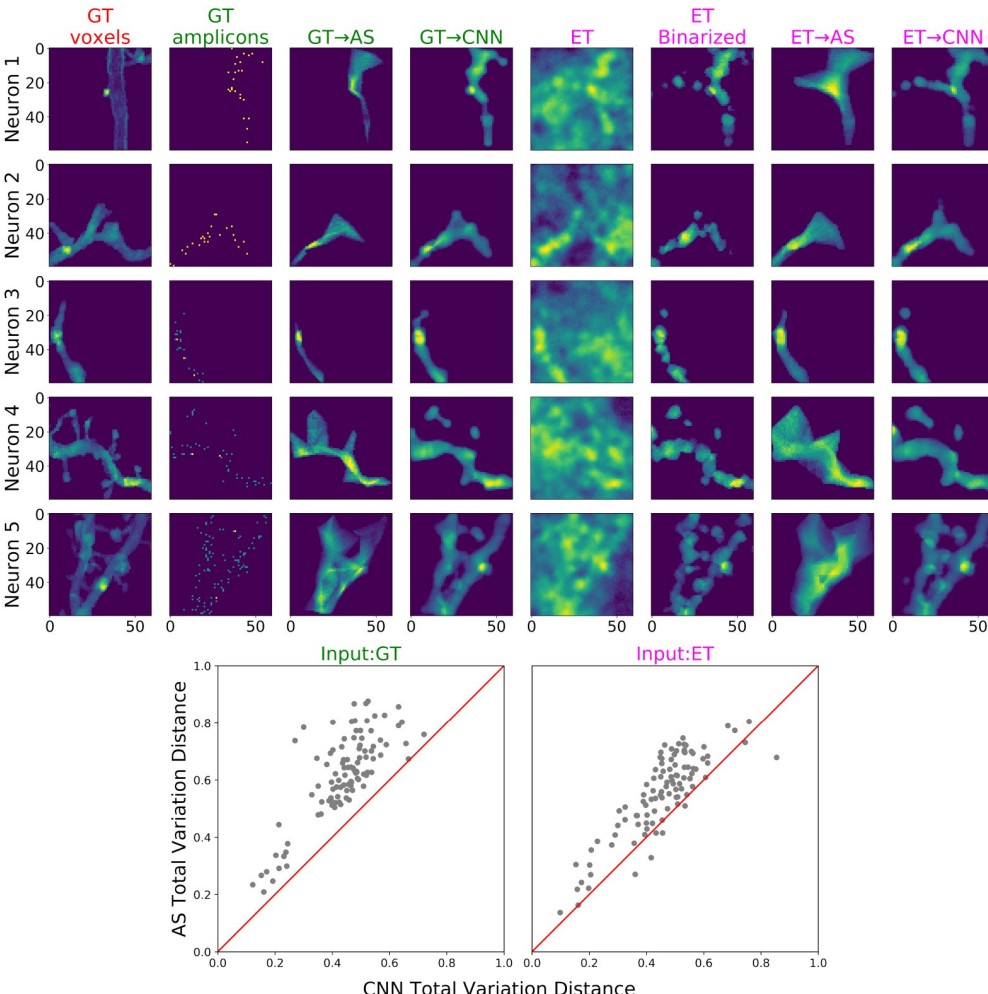

**Fig 9. Comparing the morphology reconstruction for high-resolution, dense-expression simulations, using alphashape (AS) and CNN, with either ground-truth (GT) amplicons or evidence tensor (ET) as input. Top**: Morphology prediction results for 5 examples. All the columns show the sum-projections across z-planes. The first column shows the ground-truth morphology. The next three columns show the prediction results using alphashape or CNN, when the ground-truth amplicons are used as input (indicated in green titles). The following four columns show the prediction results using alphashape or CNN, when the evidence tensors are used as input (indicated in magenta titles). Evidence tensors are first calculated using the correlation method described in S1A.3.1 Appendix, and before input into alphashape, they are binarized with threshold 0.7, as shown (again, see S1 Fig for details on the choice of this threshold). **Bottom**: Total variation distance (see S1A.3.2 Appendix) on 100 testing examples, including the five shown on the top panel, comparing the results from alphashape prediction on the input (y-axis) and CNN prediction on the input (x-axis). Each dot represents one testing example. The red lines indicate where the two methods have the same performance. Lower values are better; the CNN significantly outperforms the alphashape reconstructions here. A video of the reconstruction result (across z-planes) can be found at this link.

signals. The second approach uses Convolutional Neural Networks (CNNs) applied to the evidence tensor. This method outperforms alphashape, as shown in Fig 9.

In summary, morphological tracing is difficult with low-density STIBEs, but with sufficient density it is feasible to achieve sub-micron resolution. A thresholded version of the evidence tensor creates a satisfactory summary of the morphology, but it takes the form of a collection of discontiguous regions. To incorporate our knowledge that neurons occupy contiguous regions in space, we can use alphashape (which incorporates this knowledge directly) or CNNs (which learn from data); CNNs yield the best estimates.

## 4.2 Low-resolution, sparse-expression simulations

Next we investigate whether data from lower-resolution optical systems can still be used to reconstruct neuronal morphology. Here, we assume the optical system is less sensitive: the resolution is lower and the PSF is larger. We also assume the viral infection rate is lower, infecting only 1% of all the axonal segments and discarding all dendritic segments (emulating barcode injection at one site and axonal tracing at another site). This lower infection rate more closely mimics the low labeling efficiency by the current viral barcoding technologies. We assume amplicons can be produced with a density that is roughly consistent with the current experimental technologies, namely 0.08 amplicons per micron length [5]. More details on the low-resolution simulation can be found in S1A.1 Appendix.

The left column of Fig 10 illustrates the difficulties in analyzing this kind of STIBE. The low optical resolution and high amplicon density often cause multiple amplicons belonging to different axons to occupy the same voxel. The original colors of the target barcodes blend with the other barcodes located nearby in the original physical space, making it difficult to accurately estimate the barcodes associated with cells in a given region of tissue. (Compare this figure with the high-resolution data shown in the middle panel of the top plot of Fig 8, where colors for individual axon targets can be clearly seen).

**4.2.1 Barcode estimation.**   For this dataset, first we applied the same naïve barcode discovery approach that we used for the high-resolution simulations. This naïve method was able to find less than 40% of all the barcodes in the imagestack (assuming a maximum false discovery of 3 barcodes). Therefore, we adopted the iterative approach that was used in high-resolution high-density regime. As shown in the bottom plot of the left panel of Fig 10, four iterations of this approach were sufficient to find all the barcodes (412 in total) for this data, achieving 100% discovery rate with only 1 false positive. This result suggests that even in this more challenging low-resolution regime, an iterative approach can nonetheless recover the barcode library accurately.

**4.2.2 Amplicon estimation and morphological reconstruction.**   Neuronal tracing on this low-resolution simulation was performed using a similar approach to that taken on the high-resolution dataset. For this dataset, as shown in the top panel of Fig 11, the original evidence tensor already recovers the true amplicon locations quite accurately; a thresholded evidence tensor yields an adequate (though discontiguous) representation of the path of each neuron (see S1 Fig for details on the choice of this threshold). Alphashape or CNN methods can be used to further improve this result, giving reasonable estimates for the contiguous space occupied by each neuron (see Fig 11). All together we see that we are still able to detect the correct morphology of the captured neurons, even for this low-resolution STIBE.

## 5 Conclusion

In this work we developed detailed simulations to test the feasibility of using Spatial Transcriptomics-based Infinite-color Brainbow Experiments (STIBEs) for morphological reconstruction

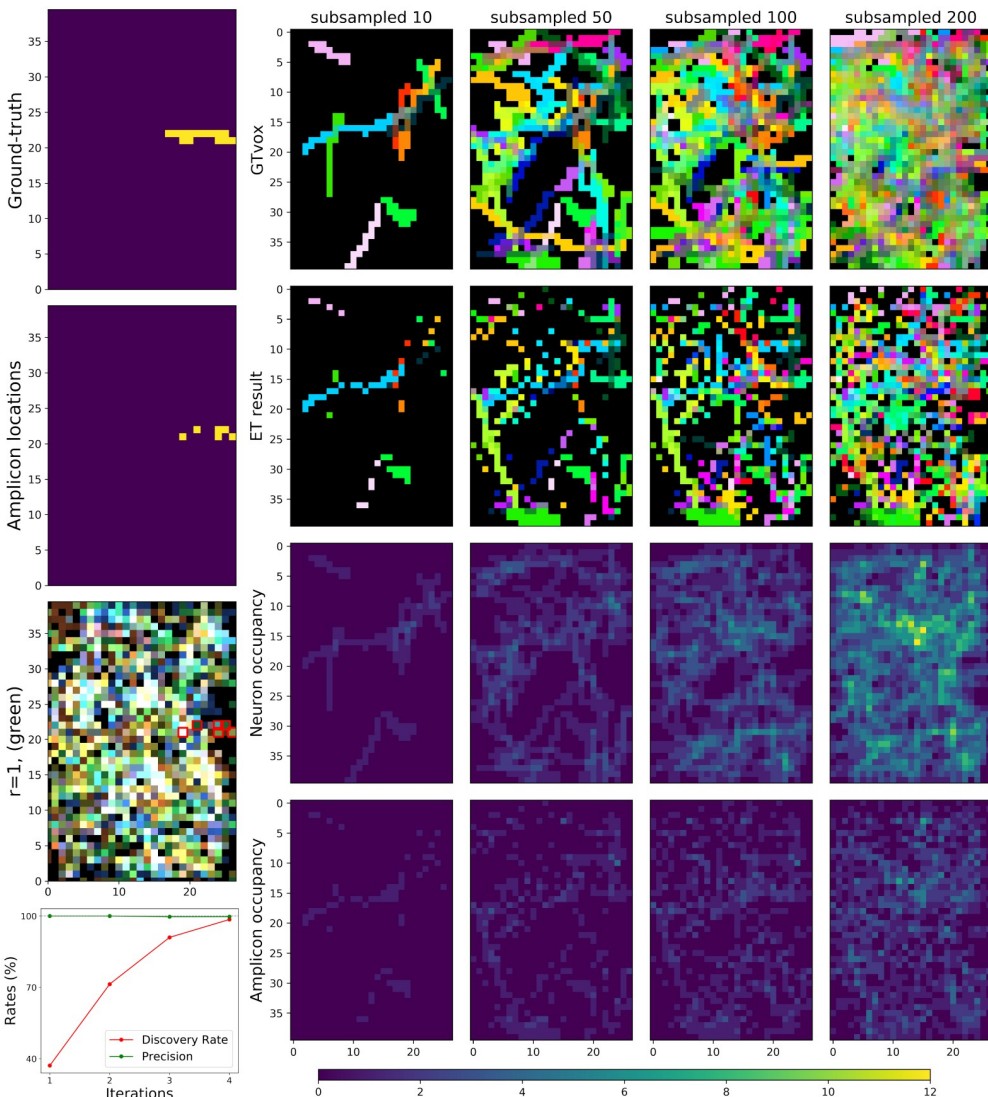

**Fig 10. Limited optical resolution in low-resolution simulations yields signal mixing between neighboring neurons, but barcode discovery is still possible through an iterative process. Left, from the first to the third row:** We show an example of one neuron. From the first to the third row: ground-truth morphology of the neuron; simulated amplicons; the image slice at the first imaging round. The red squares on the third row highlight true amplicon locations for a selected neuron, corresponding to the second row. Green fluorescence signal would be expected for the voxels containing amplicons with this neuron's barcode in the first imaging round, but other colors are also present in these voxels due to contributions from other neurons. **Bottom Left**: The Discovery Rates (DR) result of the iterative barcode discovery process described in S1A.2.2 Appendix. As we proceed through more iterations, the precision remains essentially 100% but many new correct barcodes are discovered. Four iterations were sufficient to find all the barcodes in this low-resolution simulations data. **Right**: Details of the simulated low-resolution data. The neurons are highly overlapped in this data, so for visualization purposes we here show the results after subsampling the 412 neurons present in the data. Each column corresponds to a different number of subsampled neurons, and the number of randomly selected neurons are indicated in the title of each column. The first two rows show the ground-truth neuron segments and the evidence tensor (thresholded at 0.1, see S1 Fig for details on the choice of this threshold) for the same neurons. As in Fig 8, each neuron segment is plotted with a distinct color. Note that the evidence tensor roughly captures the neuronal morphology shown in the first row. The last two rows show the number of overlapping ground-truth neuron segments and the number of amplicons for these neurons for each voxel; note the high degree of overlap.

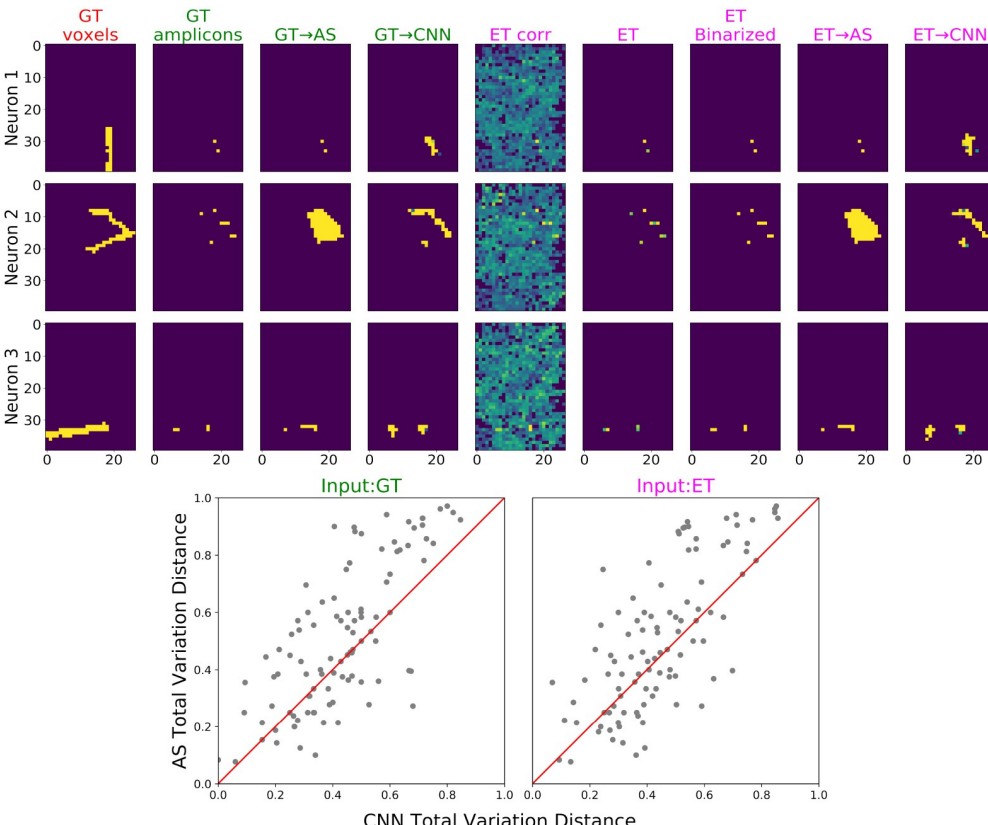

**Fig 11. Morphology reconstruction results from CNN on low-resolution, sparse-expression simulations.** Similar to Fig 9 for high-resolution simulation, this figure compares morphology reconstructions from low-resolution data using alphashape (AS) and CNN, with ground-truth (GT) amplicons and evidence tensor (ET) as input. Overall, we see that micron-precise morphological reconstructions are possible, even from low-resolution STIBEs. **Top**: Morphology reconstruction results, using either ground-truth amplicons or the evidence tensor as the input to alphashape or CNN. As in Fig 9 the colors of column titles indicate the input categories; from left to right the columns are: the ground-truth morphology, ground-truth amplicons, alphashape and CNN prediction provided ground-truth amplicons as input, evidence tensors using correlation ('corr') methods, evidence tensors using amplicon density (described in S1A.3.1 Appendix; note that this is much more accurate than the simple correlation-based method here), thresholded evidence tensor (using amplicon density thresholded at 0.1, see S1 Fig for details on the choice of this threshold), and alphashape and CNN prediction given evidence tensor as input. For the alphashape input, the evidence tensor obtained from amplicon density was binarized with threshold 0.1. **Bottom**: Evaluating the morphology reconstruction using total variation distance loss (see S1A.3.2 Appendix). On the left we show performance using ground-truth amplicon locations (GT, left) and on the right we show performance using the evidence tensor (amplicon density) derived from the imagestack (ET, right); conventions as in Fig 9. CNNs outperform alphashape modestly here.

of many neurons simultaneously. We developed a novel blind-demixing algorithm, followed by a neural network based reconstruction approach, which together achieve high barcode detection rates and accurate morphological reconstructions, even in relatively low-resolution settings.

There may be room to improve further on the methods proposed here, on at least two fronts. In the first stage, we use an iterative approach to detect barcodes: we model the presence of barcodes that are already in our library, and search the residual images for new barcodes to add to our library. We found that the proposed linear programming approach (building on previous work described in [16]) was effective for decomposing the observed signal into the sum of these two parts (i.e., the signal that can be explained by barcodes we already

know and the left over residual), but in future work it may be possible to improve further, using e.g. neural networks trained to decompose images using a simulator similar to the one used here [50].

Second, given an estimated barcode library, we need methods for recovering the morphology corresponding to each barcode from the imagestack. One promising direction would be to use more sophisticated approaches for 3D neuronal recovery, adapting architectures and loss functions that have proven useful in the electron microscopy image processing literature [51, 52]. We hope to explore these directions in future work.

Finally, we should emphasize that this study focused on a small region from a mouse brain, densely reconstructed with electron microscopy. Extending these methods to a whole-brain scale seems feasible but will require significant effort, involving stitching and registering data across many spatial subvolumes analyzed in parallel. As experimental methods continue to march forward and expand in scale [36] we hope to tackle these computational scaling issues in parallel.

## Supporting information

**S1 Appendix. Detailed methods.**
(PDF)

**S1 Fig. Finding the threshold for the evidence tensor, for high (top) and low (bottom) resolution data.** In order to create input data for the morphology prediction in Figs 8–11, we needed to binarize the evidence tensors. Here we show the test cases (one row shows one example neuron) where we used various thresholds to binarize the evidence tensors (the third columns and the following) and compared the result with both original ground-truth voxels (first columns), as well as the original (continuous) evidence tensor (second columns). For the high resolution simulation, we used 0.7 as the threshold since the result was robust. This threshold was used to visualize Figs 8 and 9. For the low resolution simulation, we used BarDensr to estimate the evidence tensor and the results were much sparser than the high resolution case, and therefore the binarization was relatively robust to the choice of the threshold. Here we used 0.1 for the low resolution case. This threshold was used to visualize Figs 10 and 11.
(TIFF)

**S2 Fig. Missed barcodes in the real data (Fig 6).** Here we show two examples of the amplicons that are from the barcodes that were not found in our amplicon detection process in Fig 6). For both panels, the red frames indicate where the correct barcode signals are expected. The left panel shows the missed barcode which had the most abundant amplicon. We see that the signal intensity varies significantly (e.g., round 2 has much higher signal compared to round 4 and 5). The right panel showed missed barcodes with the third most abundant amplicons. From round 2, 4 and 7 we see there is a relatively high phasing and/or color-mixing happening in the channel 1, which might have made the barcode discovery difficult.
(TIFF)

## Acknowledgments

We thank Xiaoyin Chen for sharing the data and helping with the analysis. We thank Li Yuan, Tony Zador, and Abbas Rizvi for many helpful discussions.

## Author Contributions

**Conceptualization:** Shuonan Chen, Jackson Loper, Liam Paninski.

**Data curation:** Shuonan Chen, Jackson Loper, Pengcheng Zhou.

**Formal analysis:** Shuonan Chen, Jackson Loper.

**Funding acquisition:** Liam Paninski.

**Project administration:** Liam Paninski.

**Software:** Shuonan Chen, Jackson Loper.

**Supervision:** Liam Paninski.

**Writing – original draft:** Shuonan Chen, Jackson Loper, Liam Paninski.

**Writing – review & editing:** Shuonan Chen, Jackson Loper, Pengcheng Zhou, Liam Paninski.

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
