## [Decision Letter · Decision Letter 0]

2 Nov 2021

Dear Chen,

Thank you very much for submitting your manuscript "Improved blind demixing methods for recovering dense neuronal morphology from barcode imaging data" for consideration at PLOS Computational Biology.

As with all papers reviewed by the journal, your manuscript was reviewed by members of the editorial board and by several independent reviewers. In light of the reviews (below this email), we would like to invite the resubmission of a significantly-revised version that takes into account the reviewers' comments.

I am very sorry for the delay with this manuscript. Fortunately, two experts with excellent match for your manuscript have reviewed the work. They have given good recommendations to better present your work. Importantly, the reviewers both identified issues with testing the method and demonstrating that the method works that need to be addressed.

We cannot make any decision about publication until we have seen the revised manuscript and your response to the reviewers' comments. Your revised manuscript is also likely to be sent to reviewers for further evaluation.

Sincerely,

Hermann Cuntz

Associate Editor

PLOS Computational Biology

Daniele Marinazzo

Deputy Editor

PLOS Computational Biology

I am very sorry for the delay with this manuscript. Fortunately, two experts with excellent match for your manuscript have reviewed the work. They have given good recommendations to better present your work. Importantly, the reviewers both identified issues with testing the method and demonstrating that the method works that need to be addressed.

Reviewer's Responses to Questions

**Comments to the Authors:**

Reviewer #1: Barcoding methods attempt to assign unique genetic barcodes to each cell. Neurons express multiple copies of those unique barcodes throughout their arbors. In-situ single-cell sequencing, which can sequence the barcodes without dissociating the tissue, may reveal outlines of multiple neuronal morphologies and the characteristics of their apposition. The present paper studies the feasibility of such experiments via computer simulations (based on existing electron microscopy-based dense segmentations) and presents a new algorithm to automate the reconstruction.

The paper is most closely related to the authors’ previous contribution Ref. [16]. This paper extends and applies Ref. [16] to reconstruction of multiple neuronal arbors in barcoded tissues, which is an important problem. While the paper offers a new computational perspective, and I’d support its publication in principle, I think multiple aspects need major revisions before it can be considered suitable for publication:

- I find the statements on the spatial extent (projection pattern in the abstract, etc) a bit overreaching. The authors actually use a careful language (e.g., “suggest the possibility”). However, the paper does not at all concern itself with scaling in space. Why should we think that this algorithm would work when the volume is millions of times larger, which is what’s needed for those claims? If the approach is to divide-and-conquer, how could we merge the subvolumes? (e.g., gluing the interface may not be straightforward.) I find the lack of at least a careful discussion of questions like these an important omission.

- The results feel somewhat anecdotal and appear to be based on a single simulation. Could the authors characterize the uncertainty by reporting e.g., mean & st dev values over a few simulation runs? Easiest could be to use the same EM volume while simulating the barcode assignments from scratch. Also, using more than one EM volume might not be hard because multiple such volumes were published in the past few years.

- I don’t think the paper's TV loss implementation is a common choice for similar computer vision tasks. Please better explain this choice and/or provide references.

- “In all cases we chose thresholds to ensure at most 3 false positives.” I find the y-axes being absolute counts somewhat inappropriate for the claims of this paper. Could you please present them as false positive rates? (e.g., count/volume, count/neuron) Also, I couldn’t understand if this rate is supposed to be representative or minimal - please explain how you chose 3. Lastly, the definition of the threshold is not clear at this point in the main text.

- The main text seems to target biologists and defers all technical details to the Methods section. This style is somewhat justified in the presence of Ref. [16]. However, as a stand-alone paper, it does not adequately communicate the technical context for a comp. bio. paper: (a) The main idea behind the algorithm of Ref. [16] could be described in more detail in the main text. (b) The differences (add-ons) of the present paper with the algorithm of Ref. [16] should be stated in the main text. (c) How does this paper relate to/compare with the group’s previous approach on a similar subject; Sumbul, …, Paninski, “Automated scalable segmentation of neurons from multispectral images”, 2016?

- Similar to the previous comment, a central claim is the “novel blind-demixing algorithm”, but this is barely explained in the main text. Again, while I somewhat understand the authors’ intention, I think the main text could do more in this direction. (e.g., what’s novel about it? what’s better about it? what makes it suitable for the problem?)

- Fig. 8 & Fig. 10: GT and CNN recons. differ quite a bit. Why is the presented accuracy good enough? Would they look much more similar if a slab (a few slices) were projected? As is, I do not get the sense that whole neuronal morphologies can be recovered faithfully. Since this is a central claim, presentation and interpretation of the results should be much more careful.

- Including a few representative BARseq slices or referring to specific figures in other papers could make it easier to visually assess the level of realism of the simulations.

- Fig. 3, right, appears saturated.

- line 238: As shown in the *bottomn* …

- line 378: \\tilde{B}’  \\tilde{B} (Please check the prime.)

Reviewer #2: In their manuscript "Improved blind demixing methods for recovering dense neuronal morphology from barcode imaging data", the authors describe simulation experiments to test the performance of their method to reconstruct dense neural morphologies from low- and medium-resolution barcode imaging data acquired with standard light microscopy.

The reconstruction method uses Spatial Transcriptomics-based Infinite-color Brainbow Experiments (STIBE). In this setup, cells express unique random barcode sequences consisting of a small vocabulary of letters (C = 4 here) that are imaged as colors. The barcodes clump in amplicons that occur at random locations and tunable frequency in each cell and are consistent across the cell. The barcode can be recovered by sequential imaging and, if the sequence is sufficiently long (R = 17 here), the chance of accidential collisions is near zero.

Reconstructing neural morphology in dense neural tissue is complicated because neurons are thinner than the resolution of light microscopy such that many voxels are occupied by many neurons, and higher resolution methods like electron microscopy (EM) are slow, expensive, and generate very large datasets for small volumes.

This paper addresses two issues:

1. Many voxels are occupied by more than one neuron, so barcodes mix and must be unmixed.

2. Dense neural morphologies must be recovered from sparse amplicon reconstructions.

The authors simulate data to test reconstruction performance under two resolution regimes, high-resolution (100nm/voxel isotropic) and low-resolution (5x5x80um/voxel) from dense neuron morphologies reconstructed from EM. In the high-resolution experiment, all neurons are labeled, and 8 amplicon densities (1-200 per um^3) are explored. In the low-resolution experiment, only sparse susbsets of axons are labeled at a fix amplicon frequency of 0.08 per um length. Image data is simulated with some ad-hoc noise and smoothing applied to the labels coming from amplicon locations.

The authors propose a simple and elegant solution to overcome that the library of barcodes present in an images is unknown. They focus on voxels that are occupied by only a single neuron first. Since the barcodes consists of only a one label in each round, voxels that have one clear label in all rounds are occupied by just one neuron. Those barcodes are added to the library, then their signal contribution is removed from the image, leaving only barcodes that have not yet been added to the library. This demixing procedure iteratively removes barcodes from the image and makes it sparser such that the remaining barcodes can be discovered and subsequently removed. The authors report discovery rates and false positive counts under various density regimes.

For shape recovery, the authors compare alphashape against a CNN trained on sparsely subsampled neuron morphologies from their example volumes. For high-resolution experiments they used a 3D network, for the low resolution experiment, they used a 2D network. The CNN recovers morphologies better than alphashape in the high-resolution experiment. In the low-resolution experiment, the difference is not as clear.

Summary:

This is a well written paper, describing a simple and elegant solution to recover neural morphologies from densely labeled barcode images at low resolution (compared to the morphology). The performance of this solution is studied on appropriate simulated data that captures a number of real world properties of expected real experiments. Transfer into real world settings will require additional work but this is not subject of this manuscript. The work presented in this manuscript is important for ongoing studies to recover connectivity from complete brains of large animals. Connectivity reconstruction as such is not addressed, but morphology reconstruction is a first and important step towards this goal. MIT licensed open source code is available at https://github.com/jacksonloper/bardensr.

While the paper is well written and easy to understand, I suggest some work on the edges to make it more valuable for readers before publication.

I have only one major request:

If I understand correctly (the description is a bit short), the CNNs for shape recovery have been trained on the same data that was used for the experiments. That means that it could have learned the shapes of the neurons that it later aims to recover. If that is true, this experiment should be repeated with training on data that is not used for the demixing experiment, either from a different part of the volume or from neurons that are then removed from the dataset.

Minor:

11-12 Either or makes no sense here, low resolution limits the number more rather than less

47 I don't understand why high density in the high-resolution regime makes reconstruction harder. Can you explain? In the low-resolution it is obvious, because all voxels have arbitrarily mixed signals, so the iterative unmixing cannot start anywhere, and fine processes would remain mixed after many iterations.

Figure 4 (and others) the GT neuron morphologies are rendered as if each voxel were occupied by only one neuron. You make the point that this is not the case which makes sense because neurons have thin processes and the reconstruction comes from high resolution EM. Can you render them by mixing the colors according to voxel coverage? You could do that by voxelizing them at e.g. 4x the target resolution, then downsample with are aaveraging. That would make the isseu more obvious.

Figure 4 The amplicon locations in the bottom panels are consistently between voxels instead of in voxel centers. The Gaussian amplicon spots look as if they are centered at voxels, not between voxels. I do not understand what that means and also do not understand how that leads to color mixing of up to 6 neurons. Can you please make this more clear?

Figure 5 Please use non-stacked mulit-histogram plots for the first two plots. This will make the hidden results in the second plot visible and the iterations axis is not continuous anyways.

Figure 7 Why are the recovered amplicons so big? Wouldn't it be better to detect their center points? Or are they actually that big and so they can fit only where the neuron has a caliber of about 1um? Cna you discuss this briefly?

Figure 8 May be 3D renderings of the shapes could be more informative but not sure. Slices are often misleading about the true nature of differences because 3D context is missing. But again, you may have tried this and it was worse? Also, again, why did you choose to threshold the fuzzy amplicon images instead of localizing them? Does it work better?

234--241 The entire section 4.2.1 is written as if the paper before didn't really exist and is confusing, can you please revisit this? May be giving the naive approach and the iterative approach a name, and not call the same iterative method "an iterative approach" but "the iterative approach"?

264 "like improved methods" is probably a typo

Figure 9 Like in Figure 4, it would be much clearer to see how neurons partially occupy individual voxels. Probably more important here than in the high-resolution experiment. Can you voxelize at a higher resolution and then resize with averaging?

Figure 9 The red boxes in the simulated image are practically invisible, may be put a thicker frame around them?

Figure 10 How did you decide that the evidence tensor should be thresholded at 0.1 (low-resolution) and 0.7 (high-resolution), respectively?

Algorithm 1 4 The indices r,c are not used inthe foreach loop and can be removed.

7, 8 none of the indices are used and so they can be removed

Algorithm 2 index j is not used and can be removed

A.1.1 and A.1.2 As in Figures 4 and Figure 9 I suggest to voxelize at much higher resolution, then downsample with area averaging to mix colors (or mix colors maively)?

Algorithm 3 each iteration is affected by noise and I would expect that noise to accumulate with more iterations. Is that a valid concern? May be not important with low iteration counts as in these experiments.

**Have the authors made all data and (if applicable) computational code underlying the findings in their manuscript fully available?**

Reviewer #1: Yes

Reviewer #2: **No: **MIT licensed open source code is available at https://github.com/jacksonloper/bardensr but I have not found any data (e.g. the voxalized volumes or the meshes)

PLOS authors have the option to publish the peer review history of their article (what does this mean?). If published, this will include your full peer review and any attached files.

Reviewer #1: No

Reviewer #2: **Yes: **Stephan Saalfeld
---

## [Decision Letter · Decision Letter 1]

7 Mar 2022

Dear Chen,

We are pleased to inform you that your manuscript 'Blind demixing methods for recovering dense neuronal morphology from barcode imaging data' has been provisionally accepted for publication in PLOS Computational Biology.

Best regards,

Hermann Cuntz

Associate Editor

PLOS Computational Biology

Daniele Marinazzo

Deputy Editor

PLOS Computational Biology

Please consider the suggestions by Reviewer #2 to improve the final versions of the figures.

Reviewer's Responses to Questions

**Comments to the Authors:**

Reviewer #1: The authors have adequately addressed my concerns.

Reviewer #2: I find the paper improved. Please consider the below suggestions:

Figure 3 Color mixing helps, but I find the way the colors mix surprising and unexpected. What color model do you use to mix? Your background is black, so RGB additive should be fine and give the desired result (moving the figure away from your eyes will make it disappear). Anything more fancy, in this context, is just confusing (even with good intentions). In understand that you do not want to mix with background to indicate that there is only one neuron present, however, it may make sense to treat this the same because it isn't a complete pixel. Your choice.

The right panel (100 amplicons/um^3) is essentially noise free. Compression artifact or intended?

Figure 4 The colors in overlapping pixels in the left two panels are still not mixed and therefore inconsistent with the partial overlap map on the right. I do not understand why this is handled differently than the other figures. The pixels in the left bottom panel are offset by 0.5 pixels which does not correspond with the pixels in the right panel and therefore the cross markings of amplicon locations in the plane are, again, at pixel corners.

Figure 5 This is better. Fonts are too small though and additional spacing between plots is required to help associate the axis labels with the corresponding axes. PLease zoom in the middle panel y-axis to [99:100]%.

**Have the authors made all data and (if applicable) computational code underlying the findings in their manuscript fully available?**

Reviewer #1: Yes

Reviewer #2: **No: **I was not able to find the experimental data in the repository https://github.com/jacksonloper/bardensr

PLOS authors have the option to publish the peer review history of their article (what does this mean?). If published, this will include your full peer review and any attached files.

Reviewer #1: No

Reviewer #2: **Yes: **Stephan Saalfeld

---

## [Editor Report · Acceptance letter]

31 Mar 2022

PCOMPBIOL-D-21-01522R1 

Blind demixing methods for recovering dense neuronal morphology from barcode imaging data

Dear Dr Chen,

I am pleased to inform you that your manuscript has been formally accepted for publication in PLOS Computational Biology. Your manuscript is now with our production department and you will be notified of the publication date in due course.

With kind regards,

Olena Szabo
